# Tracing Autism Traits in Large Multiplex Families to Identify Endophenotypes of the Broader Autism Phenotype

**DOI:** 10.3390/ijms21217965

**Published:** 2020-10-27

**Authors:** Krysta J. Trevis, Natasha J. Brown, Cherie C. Green, Paul J. Lockhart, Tarishi Desai, Tanya Vick, Vicki Anderson, Emmanuel P. K. Pua, Melanie Bahlo, Martin B. Delatycki, Ingrid E. Scheffer, Sarah J. Wilson

**Affiliations:** 1Department of Medicine, Austin Health, The University of Melbourne, Heidelberg, VIC 3084, Australia; kjtrevis@gmail.com (K.J.T.); c.green@latrobe.edu.au (C.C.G.); tdesai@unimelb.edu.au (T.D.); pk.pua@unimelb.edu.au (E.P.K.P.); i.scheffer@unimelb.edu.au (I.E.S.); 2Melbourne School of Psychological Sciences, The University of Melbourne, Parkville, VIC 3010, Australia; vaa@unimelb.edu.au; 3Victorian Clinical Genetics Services, Murdoch Children’s Research Institute, Parkville, VIC 3052, Australia; natasha.brown@vcgs.org.au (N.J.B.); martin.delatycki@vcgs.org.au (M.B.D.); 4Barwon Health, Geelong, VIC 3220, Australia; cats-info@unimelb.edu.au; 5Department of Psychology and Counselling, School of Psychology and Public Health, La Trobe University, Bundoora, VIC 3086, Australia; 6Bruce Lefroy Centre for Genetic Health Research, Murdoch Children’s Research Institute, Parkville, VIC 3052, Australia; paul.lockhart@mcri.edu.au; 7Department of Paediatrics, The University of Melbourne, Parkville, VIC 3010, Australia; 8Psychological Service, The Royal Children’s Hospital, Parkville, VIC 3052, Australia; 9Clinical Sciences Research, Murdoch Children’s Research Institute, Parkville, VIC 3052, Australia; 10Population Health and Immunity Division, The Walter and Eliza Hall Institute of Medical Research, Parkville, VIC 3052, Australia; bahlo@wehi.edu.au; 11Department of Medical Biology, The University of Melbourne, Parkville, VIC 3010, Australia; 12The Florey Institute of Neuroscience and Mental Health, Parkville, VIC 3052, Australia

**Keywords:** Broader Autism Phenotype, genetic, autism spectrum disorder, multiplex family

## Abstract

Families comprising many individuals with Autism Spectrum Disorders (ASD) may carry a dominant predisposing mutation. We implemented rigorous phenotyping of the “Broader Autism Phenotype” (BAP) in large multiplex ASD families using a novel endophenotype approach for the identification and characterisation of distinct BAP endophenotypes. We evaluated ASD/BAP features using standardised tests and a semi-structured interview to assess social, intellectual, executive and adaptive functioning in 110 individuals, including two large multiplex families (Family A: 30; Family B: 35) and an independent sample of small families (*n* = 45). Our protocol identified four distinct psychological endophenotypes of the BAP that were evident across these independent samples, and showed high sensitivity (97%) and specificity (82%) for individuals classified with the BAP. Patterns of inheritance of identified endophenotypes varied between the two large multiplex families, supporting their utility for identifying genes in ASD.

## 1. Introduction

Autism Spectrum Disorders (ASD) are a group of neurodevelopmental conditions characterised by deficits in social communication and social interaction, and restricted and repetitive patterns of behaviours, interests, or activities [1]. Recent estimates from the Centers for Disease Control and Prevention indicate a prevalence of 1 in 54 children aged 8 years for ASD [2]. There is strong evidence for the contribution of genetic factors to the aetiology of ASD [3]. Despite recent molecular advances, the cause remains unidentified in the majority of cases [4].

The recognition of milder phenotypes has facilitated an increased understanding and interest in the wide spectrum of clinical presentations in ASD. Family members of individuals with ASD have been observed to express milder forms, known as the Broader Autism Phenotype (BAP), that are not sufficient to meet diagnostic criteria for ASD [5,6]. The BAP includes a range of subtle behavioural and cognitive features that have qualitatively similar presentations to the core domains of ASD. While clinically significant impairment in these areas of functioning is seen in ASD [1,7], BAP traits are continuously distributed in the general population [8]. Several studies have reported the expression of at least one BAP trait in 20% of relatives of children with ASD, as well as a higher rate of BAP traits compared to controls across domains of pragmatic language [9,10], personality [11], social cognition [10,12,13] and executive function [14,15]. Monozygotic twins demonstrate 65–90% concordance for ASD [16,17] with a higher estimate when the BAP is considered [18,19], while dizygotic twin concordance is ~20% [20]. Together these findings suggest a strong genetic basis for ASD.

An estimated 25% of ASD cases can be identified clinically or molecularly with a predominant monogenic cause [21,22,23]. Although the aetiology for the majority of cases is still unknown, increasing evidence suggests a polygenic basis due to the interaction of multiple genetic risk variants following complex inheritance, with or without environmental factors [24,25,26,27]. Complementary techniques will be necessary to investigate the genetic aetiology of ASD in unsolved cases. Typically, family studies combine many small families (2–3 affected individuals), however, these are likely to be confounded by genetic heterogeneity. Very large multiplex families (> 8 affected) where ASD traits appear dominantly inherited are rare, but more genetically homogeneous. In other complex disorders, such as epilepsy, phenotypic characterisation of such families has proved powerful in gene discovery [28], however this approach has received limited attention in ASD. In multiplex ASD families, the identification of family members with BAP traits, or endophenotypes, may serve as markers of risk variants for ASD [29,30]. In turn, this may facilitate gene identification [31].

Endophenotypes are measurable features within a disorder that are proposed to reduce its complexity into more quantifiable elements [32]. These components can be represented at any level of analysis, including but not limited to biochemical, neuroanatomical, neurophysiological, cognitive or neuropsychological measurements. They have been hypothesised to reflect more aetiologically homogeneous subgroups within genetically heterogeneous conditions. Endophenotypes are also presumed to be located closer to aetiological mechanisms in the pathway between genotype and disease, compared to more overt phenotypes that are used to define clinical syndromes [33,34]. Clinically, the use of endophenotypes offers increased statistical power to localise and identify genes associated with disease [35]. Importantly, an endophenotype must indicate genetic susceptibility to disease independent of disease status, and by definition may serve as a marker of genetic liability in individuals without the disorder [34]. In individuals with the BAP, the mild expression of ASD-related traits is hypothesised to be due to an increased genetic liability for ASD. There are several BAP traits that may be considered “endophenotypes” from within the domains of language, executive function, and social cognition [31]. In the context of a single large family where numerous individuals demonstrate ASD or the BAP, recognition of BAP endophenotypes should allow granular identification of autism genes of dominant effect. This study is the first known to the authors to apply this approach in autism.

The aim of our study was to analyse ASD traits within large multiplex families to examine inheritance of ASD by identifying endophenotypes of the BAP. We achieved this aim through an iterative process of characterising, refining, and assessing BAP endophenotypes. This involved (1) rigorous phenotyping of large multiplex families to identify BAP traits and potential endophenotypes, (2) validation of putative endophenotypes and their cut-off scores in an independent aggregated sample of 20 small families (each with at least one member with ASD), and (3) classification of BAP endophenotypes in large multiplex families (from step 1) based on optimal cut-off scores following validation analysis (from step 2) to assess their utility for examining inheritance patterns. We hypothesised that (1) multiple individuals in large families would demonstrate the BAP, (2) distinct BAP endophenotypes would be identified, and (3) BAP endophenotypes would vary in presentation between large multiplex families.

## 2. Results

### 2.1. Hypothesis 1: Multiple Individuals in Large Families Demonstrate the BAP

Based on our rigorous protocol for phenotyping the BAP in large multiplex families, we identified 32 members with the BAP across Family A and B. Of the 23 members in Family A who did not have an ASD diagnosis, we detected the BAP in 17 (74%) individuals, with 6 individuals unaffected. In Family B, we detected 15 (63%) individuals with the BAP, with 9 individuals unaffected (Table 1).

Intellectual function was assessed in 54/63 (86%) individuals. Overall, participants were of average or greater intelligence. Average full scale intelligence quotient (FSIQ) was observed in 32/54 (59%) individuals, while 20/54 (37%) demonstrated superior or very superior FSIQ (Table 1). We performed group-level comparisons of cognitive, social and adaptive functions between family members with and without the BAP using non-parametric and parametric tests (Mann–Whitney U and *t*-tests, respectively), with the more conservative parametric tests reported here as there were no differences between these approaches. On average, individuals with the BAP demonstrated poorer pragmatic language, with significantly higher mean Pragmatic Rating Scale (PRS) scores (*M* = 8.75, *SD* = 7.08) compared to unaffected individuals (*M* = 2.00, *SD* = 3.05), *t* (40.99) = −4.42, *p* < 0.001. No significant differences were observed for general intellect, the Faux Pas Task (FPT), executive (D-KEFS) or adaptive function measures (ABAS-II, BRIEF; See Materials and Methods).

### 2.2. Hypothesis 2: Specific BAP Endophenotypes Exist Across BAP Domains

Based on our iterative characterisation process, four distinct endophenotypes of the BAP were reliably identified in the small families sample. From the natural grouping of traits, these reflected “socially unaware”, “pedantic”, “aloof”, and “obsessive’” endophenotypes (Table 2). At the highest level of the dendrogram of the 33 BAP traits there was a clear split, whereby traits of the socially unaware and pedantic endophenotypes were more similar to each other and more dissimilar to the combination of traits of the aloof and obsessive endophenotypes. There was a significant difference between the mean proportional scores of the unaffected and BAP groups, with the BAP group demonstrating significantly higher scores on all four endophenotypes (all *p* < 0.015).

Analysis of receiver operating characteristic (ROC) curves indicated relatively good discrimination within the small families for the socially unaware, aloof and obsessive endophenotypes (all AUC > 0.73, all *p* < 0.025), and acceptable discrimination for the pedantic endophenotype (AUC = 0.68, *p* = 0.077). Although we note that Box’s M was violated in the discriminant function analysis (likely due to variation in the sample sizes), combined the four endophenotypes captured 93% of cases (Wilk’s λ = 0.47, χ^2^ = 27.83, *p* < 0.001). In particular, the endophenotypes showed high sensitivity for the BAP group (97%) characterised by higher proportional scores, and good specificity for the unaffected group (82%) with lower proportional scores (Table 2).

### 2.3. Hypothesis 3: BAP Endophenotypes Vary in Large Multiplex Families

Applying the endophenotype thresholds to the proportional scores of the 33 BAP traits for members of Family A and B led to the identification of all individuals classified as having the BAP. Two additional BAP cases were identified in Family B based on the presence of above threshold endophenotype scores, indicating good utility of this approach (Figure 1). One individual was excluded from this analysis due to incomplete data. Across both families, the aloof endophenotype was most common (62%), followed by obsessive (60%), pedantic (55%) and socially unaware (48%). Approximately one quarter of family members met criteria for only one endophenotype, 15% met criteria for two, and the remainder met criteria for 3–4 (62%) (Figure 1). The dominant endophenotype across both families, as determined by the highest score, was aloof (47%), followed by obsessive (26%), socially unaware (18%) and pedantic (9%).

Family A appeared to have two endophenotype profiles, with one characterised by the presence of a single endophenotype (35%) seen in individuals who were mostly married-in (67%), contrasting with the second profile (41%) of all four endophenotypes, most evident in core family members (72%) (Figure 1). Overall, the obsessive endophenotype occurred most frequently (77%), followed equally by pedantic (65%) and aloof (65%), and then socially unaware (53%). The co-occurrence of the obsessive and pedantic endophenotypes was relatively common, seen in 29% of married-ins and core family members. Overall, there was a range of dominant endophenotypes across individuals, with aloof the most frequent (35%) particularly in core family members (83%).

Contrasting with Family A, Family B had more individuals (70%) with multiple endophenotypes, in both married-in and core family members (Figure 1 and Figure 2). All four endophenotypes were again most frequently observed in core family members, indicative of a more severe BAP presentation. Unlike Family A, however, the aloof endophenotype occurred most frequently in Family B (88%), followed by obsessive (71%), pedantic (65%), and socially unaware (65%). The aloof endophenotype was also identified as dominant (59%), evident in 70% of core family members.

### 2.4. Correlates of the BAP Endophenotypes

Across both families, no sex or age differences were observed for any of the endophenotypes (all *p* > 0.200). Overall, a more severe BAP presentation (indicated by a greater number of endophenotypes) was associated with reduced social adaptive functioning on both self-report and objective measures of social communication (Table 3), demonstrating good convergent validity. In particular, a more severe BAP presentation showed a strong correlation with more severe pragmatic language difficulties, with scores for each endophenotype also significantly correlated. A similar relationship was evident for the ability to detect a faux pas in social discourse and self-reported social functioning, particularly for family members with the socially unaware endophenotype (Table 3).

For the cognitive measures, a more severe BAP presentation was associated with reduced executive functioning, particularly for nonverbal measures of cognitive flexibility (switching and fluency; Table 3). A pattern of weaker correlations was also evident for specific endophenotypes, including lower IQ in the socially unaware and aloof endophenotypes (Table 3).

## 3. Discussion

We investigated the BAP to capture phenotypic variation within and between high-risk ASD families, with the aim of improving characterisation and identification of individuals for accurate molecular genetic analysis. We identified multiple individuals with the BAP in large multiplex families using rigorous phenotyping and a novel reliable endophenotyping approach, validated in an independent sample of small ASD families and with objective measures of cognitive, social and adaptive functioning. This deep phenotyping showed that specific BAP endophenotypes exist beyond the conventional BAP domains of social relationships, communication, and circumscribed interests and behaviour, allowing for more granular detection of subtle features of the BAP. Distinct patterns of inheritance were identified by applying the endophenotype approach in two large multiplex families, highlighting the utility of such a framework to identify putative ASD genes of dominant effect.

The research model employed here to phenotype rare large multiplex families reveals a pattern consistent with autosomal dominant inheritance of ASD/BAP traits that would not have been captured without deep phenotyping. Fifteen individuals (23%) met criteria for ASD and 33 (51%) the BAP, including some married-in individuals. Our promising endophenotype analysis provides further insight into specific profiles of the BAP and its varied presentation. Traditionally, ASD family studies include 2–3 affected individuals [36,37]. For example, four candidate ASD genes were identified in seven ASD/BAP pedigrees with ≥ 3 affected individuals [38]. Larger multiplex families remain scarce in the literature [30,39]. Here, we identified more subtle indicators of carrier status in two large families, using a robust endophenotyping method with good sensitivity and specificity to detect the BAP in two independent samples.

The BAP is strongly associated with ASD and may be considered a marker of genes that contribute to ASD risk [31,40]. Here we delineated the BAP into distinct endophenotypes that fulfil Gottesman and Gould’s criteria for a true endophenotype (Appendix A) [41]. This includes recent proposed revisions to account for the strong overlap in the complex aetiology and genetic liability underlying the spectrum of trait expression across many neuropdevelopmental and psychiatric conditions, with the expectation that putative endophenotypes may not be strictly disorder-specific [34]. Each endophenotype cluster was characterised by a combination of communication, personality and behavioural indicators showing how specific traits across the traditional BAP domains may group together to form distinct endophenotypes or ‘profiles’ that capture phenotypic variation within and between families. As summarised in Table 4, these profiles capture identifiable ‘personas’ that have core characteristics with high face validity. These profiles also vary with functional correlates in distinct ways, supporting their construct validity. For example, the aloof endophenotype was characterised by a lack of innate social motivation or ability to meaningfully connect and empathise with others, associated with decreased theory of mind, and lower executive and intellectual functioning. One individual dominant for the aloof endophenotype described social interactions as “a means to an end”. In contrast, the pedantic endophenotype was primarily characterised by detail-oriented traits, showing no associations with intellectual, executive or adaptive functions. Unsurprisingly, given the importance of social communication deficits in ASD and the BAP, all endophenotypes were associated with poor social communication, with the socially unaware endophenotype most broadly affected across social, intellectual, executive and adaptive functions (Table 4).

In contrast to conventional approaches that commonly rely on broad and non-specific classification of the BAP, the endophenotype framework implemented in the present study reliably identifies individuals that meet threshold criteria for a specific endophenotype profile potentially linked to susceptibility genes that confer risk for ASD. Clustering traits across conventional BAP domains to achieve granular characterisation of endophenotype profiles may thus improve detection of the BAP to facilitate gene discovery. We propose that such a framework is necessary to increase the efficacy of assessment protocols. It may also allow for a more sophisticated mapping of psychological and neural correlates to delineate the neurobiology of ASD, which has been characterised by inconsistencies in the literature to date [42,43,44,45,46].

Phenotypic heterogeneity was evident in both families at the endophenotype level suggesting that a single familial mutation may produce a phenotypic spectrum, with other genetic, epigenetic and environmental factors potentially influencing expression. With the advancement of high-throughput next generation sequencing technologies, meticulous phenotypic characterisation of both affected and apparently unaffected individuals remains essential for accurate data interpretation. In other words, identification of subtle endophenotypes, such as the four identified here, are crucial for advancing gene discovery programs. Importantly, these proposed BAP endophenotypes should be replicated and validated in subsequent research and further endophenotypes sought through a deep phenotyping approach.

Although multiplex families with ASD are genetically homogeneous, our phenotyping analysis suggests possible bi-lineal inheritance of the BAP in both families. Therefore, multiple risk alleles may contribute to ASD/BAP in later generations, consistent with recent genetic and phenotyping evidence [47,48]. The importance of unique de novo genetic changes in both sporadic (or ‘simplex’) ASD [25], and small multiplex ASD families [39] has become increasingly apparent. However, with at least seven individuals with ASD and many more with the BAP in our families, there is less likelihood of de novo changes contributing to each phenotype. It is much more likely that there is a single genetic variant of major phenotypic effect in each family, with the possibility that there are additional de novo genetic changes in some individuals that contribute to phenotypic severity.

The intensive nature of the study meant that clinicians were not blinded to family relationships, potentially leading to investigator bias. However, our diagnostic method of using the independent assessments of experienced clinicians and subsequent consensus agreement aligns with current best practice for ASD/BAP diagnosis. This method was further strengthened by objective cognitive and behavioural testing using quantitative measures. We selected a relatively low threshold for BAP classification, leading to the identification of many affected individuals. However, this approach is justified in a family with a clear genetic liability for ASD and was validated by the finding of consistent data-driven endophenotypes in the small families. Successful gene identification in future work requires capture of all individuals who may carry the putative variant, with the approach outlined here designed to enable more robust gene identification work. Future genetic investigations are required to test the reproducibility of the four identified endophenotypes over time and in additional independent samples, as well as determine their rate of occurrence in affected families compared to the general population (see Appendix A for materials and guidelines on assessment and scoring of the four endophenotypes).

## 4. Materials and Methods

### 4.1. Large Multiplex Families

Large multiplex families were primarily ascertained from the Barwon Autism Database as part of a broader Collaborative Autism Study [49]. For inclusion as a multiplex family, >8 individuals with a diagnosis or suspected diagnosis of ASD or the BAP were required. The two fully characterised large multiplex families used to examine inheritance patterns using BAP endophenotypes are referred to as ‘Family A’, ascertained from the Barwon Autism Database, and ‘Family B’, who were self-referred. All available relatives were recruited, including those with and without reported BAP traits. The study was approved by the appropriate institutional review boards including the Human Research Ethics Committees of Barwon Health (HREC 02/34 and 04/57; reviewed 9 August 2017) and The Royal Children’s Hospital (HREC 25043Y; reviewed 26 September 2017), Australia. Informed consent was obtained from all participants or a parent/guardian, and all study methods were carried out in accordance with relevant guidelines and regulations.

### 4.2. Protocol for Diagnosing ASD in Large Multiplex Families

ASD diagnoses were confirmed using the Autism Diagnostic Observation Schedule—Generic [50] (ADOS-G), or the Autism Diagnostic Interview—Revised [51] (ADI-R), based on Diagnostic and Statistical Manual of Mental Disorders, Fourth Edition, Text Revision (DSM-IV-TR) criteria [52]. For adults, the structured Family History Interview [5] (FHI) was administered by NJB, while for adolescents, a detailed developmental and medical history was obtained. Quantitative measures of intellect, executive functioning, adaptive behaviour and social functioning were also completed (Table 5). Testing was undertaken over a number of days to minimise fatigue effects. A physical examination was conducted for dysmorphic and neurocutaneous features and growth parameters. Standard genetic testing (karyotype, fragile X testing) and metabolic investigations were performed on probands.

Across Family A and B, 65 individuals were recruited: 16 children (2–12 years), 9 adolescents (13–17 years) and 40 adults (18–79 years) spanning 4 generations. Of these, 16/65 met criteria for a diagnosis of ASD. Family B also reported a deceased family member who had a diagnosis of ASD (not shown on the pedigree to preserve anonymity), an additional family member with ASD who was not recruited, and a child who had been diagnosed with ASD but was not assessed. Scrambled pedigrees of affected status are presented to preserve participant anonymity (Figure 3). In each family, individuals directly related to a matriarch are classified as ‘core family’; others are referred to as ‘married-in’. Family A comprised 30 individuals, including 7 diagnosed with ASD (6/9 children, 1/6 adults; Figure 3a). Nineteen were core family; 11 were married-in. In Family B, the matriarch (II-2) was elderly and too unwell to be behaviourally assessed and subsequently passed away during the study. Three children and three adolescents participated in a limited range of phenotyping activities and as such, these individuals were excluded from final analyses. Nine participants had ASD (5/7 children, 1/5 adolescents, 3/22 adults; Figure 3b); 31 were core family, and four were married-in.

### 4.3. Protocol for Phenotyping the BAP in Large Multiplex Families

We employed a mixed methods approach to rigorously assess the BAP, including an evaluation of general intellect, executive functions, adaptive behaviour, social cognition and language pragmatics (Table 5). A purpose developed semi-structured interview, the Broader Autism Phenotype Interview (BAPI), was also administered by three clinicians with expertise in neurobehavioural disorders (NJB, SJW, IES) to all individuals ≥ 13 years to characterise the presence, nature and extent of BAP traits. Interview responses were independently rated by the clinicians on all traits, with consensus agreement used to determine the presence of the BAP. Questions focused on the participant’s life story, personal qualities, relationships, social functioning, and developmental, medical, psychiatric and vocational history. During the interview we included one or two “intentional errors” to elicit pragmatic elements of the BAP, such as terse speech [10]. Quantitative measures of social cognition and language pragmatics were also included within the interview. Social cognition was assessed using an adapted Faux Pas Task [53,54] (FPT), involving the standardised administration of four faux pas stories and four control stories [55] (maximum score = 40, *M* = 37, *SD* = 4). Pragmatic language was assessed with the Goldman-Eisler Cartoon task [56], which has previously been used to assess overly detailed speech and longer pauses between words in the BAP [10]. This task measures discourse production by eliciting a description of an eight frame captionless cartoon, “The Cowboy Story”, over three successive trials [57]. Control individuals show increased verbal fluency with successive trials compared with decreased fluency in individuals with communication deficits [56]. Following the interview, the Pragmatic Rating Scale (PRS) [10] was independently completed by the three clinicians and consensus ratings reached. A score ≥ 4 defined pragmatic impairment [11] (Table 5).

Other cognitive domains and adaptive behaviour were assessed by a member of the team trained in psychometric assessment (TV) on a separate testing occasion. This included estimating full scale (FSIQ), verbal (VIQ) and performance (PIQ) intelligence quotients, derived with the four subtest Weschler Abbreviated Scale of Intelligence [58] (WASI; *M* = 100, *SD* = 15). Executive functions were measured with seven subtests of the Delis–Kaplan Executive Function System [59] (D-KEFS; *M* = 10, *SD* = 3). The second edition of the Adaptive Behavioural Assessment System [60] (ABAS-II) and the Behavioural Rating Inventory of Executive Function [61] (BRIEF) were used to assess adaptive functioning. The quantitative assessment provided a measure of convergent validity for the BAPI. At the completion of testing, final review of all qualitative and quantitative data by the three clinicians was used to confirm BAP status based on consensus agreement.

### 4.4. Small Families

We recruited an independent sample of 45 individuals from 20 small families with at least one member diagnosed with ASD, through advertisements and from the Barwon Autism Database. All participants provided written informed consent, as described above. Inclusion criteria were: (i) no diagnosis of ASD (based on DSM-IV or DSM-V criteria), (ii) ≥1 family member with ASD (based on DSM-IV or DSM-V criteria), and (iii) >12 years of age. Individuals were classified as having the BAP if they met ≥2 criteria for a BAP diagnosis on the Broader Autism Phenotype Rating Scale [5] (BAPRS) administered by an independent ASD expert (CG). Individuals were classified as unaffected if they did not meet criteria for any BAP traits or a diagnosis of ASD. This identified 30 individuals with the BAP (4 adolescents, 26 adults) in the 20 families, ranging in age from 14–71 years, and 11 unaffected adult family members ranging in age from 18–53 years. Four adult individuals showed only one BAP trait on the BAPRS and thus, were excluded from analyses based on the above criteria. All 45 individuals were also administered the BAPI by the independent ASD expert (CG) to characterise and rate their BAP traits. To ensure inter-rater agreement, video recordings of the interviews of a subset of these individuals were independently rated by two of the three clinicians (SJW, IES), with ratings confirmed by consensus agreement.

In 31/45 individuals, the WASI-II [62] was administered to measure Verbal Comprehension (VCI) and Perceptual Reasoning (PRI) and to estimate FSIQ. All individuals were within the normal range based on FSIQ (Table 6), with no significant differences between unaffected and BAP individuals for age or intellect (all *p* > 0.250). For 27 individuals, average total scores were available for the Broader Autism Phenotype Questionnaire (BAPQ) collected as part of a separate study. Consistent with expectations, there was a trend for higher scores on the BAPQ in the BAP group, with a medium effect size (*t* (24.56) = −1.96, *p* = 0.062, *d* = 0.70).

### 4.5. Endophenotyping Procedure

We used an iterative process to characterise, refine and assess endophenotypes of the BAP in the two independent samples, as summarised in Figure 4.

#### 4.5.1. Step 1: Identification of Potential BAP Endophenotypes in Large Multiplex Families

Using a grounded theory approach, BAP traits were initially identified from a detailed literature review targeting the theoretical domains described in the seminal work of Bolton (1994), on which the conceptualisation of the BAP is largely based. The domains included speech, literacy, pragmatics, relationships, and circumscribed interests, which were explored in-depth using our BAP phenotyping protocol (described above) in members of a number of unrelated large multiplex families primarily ascertained through the Collaborative Autism Study [55]. This in-depth characterisation was phenomenologically based [63], whereby the number of traits within each domain was fully expanded through administration of the semi-structured interview (BAPI) with separate family members until no further traits were identified (saturation) to capture the entire range of BAP traits (Appendix A).

This deep phenotyping produced an exhaustive list of 36 BAP traits. Ordinal ratings of these traits were then assigned to capture subtle variations in their presentation, with severity rated on a scale of 0 = absent, 1 = mild, 2 = moderate, and 3 = severe. The presence of traits through each individual’s developmental history was also evaluated where available. Exploratory hierarchical cluster analysis was then performed to identify potential BAP endophenotypes. We used Ward’s method with Euclidean squared distances based on z-scores to progressively group traits by minimising the variability within clusters and maximising the variance between clusters [64]. Interpretation of cluster groupings was informed by the relative similarity and dissimilarity in the linkage output combined with clinical judgement, leading to the initial identification of five endophenotypes. Inspection of these endophenotypes revealed a consistent rating of 0 for two of the 36 traits across all interviews, leading to their removal. One further trait reflecting inflexibility to intentional errors was removed due to challenges in reliably assessing the trait across interviewers, resulting in a final set of 33 BAP traits (Appendix A).

#### 4.5.2. Step 2: Validation of BAP Endophenotypes in Small Families

In the small families sample, an independent expert in ASD assessment (CG) interviewed and rated 45 participants on the 33 BAP traits based on all qualitative and quantitative data, with a subset (9%) rated via consensus between CG, IES and SJW to ensure consistency in ratings across both samples and to clarify borderline cases. As above, Ward’s hierarchical cluster analysis was used to examine natural trait groupings. This led to the identification of four endophenotypes that showed a high degree of similarity to the initial five cluster solution (Spearman’s *r* range = 0.710–0.976, *p* < 0.001).

To account for a varying number of traits in each cluster we computed proportional scores, whereby scores on each trait (range 0–3) were summed and divided by the maximum total score for that cluster, to produce four cluster scores for each individual. An ROC curve was plotted for each cluster in the small families sample to identify optimum cut-off scores for determining endophenotypic status using Youden’s Index to allow mildly affected individuals to be included [65,66]. The highest score was used to represent the most prominent endophenotype for each individual, calculated as the difference between the observed endophenotype (i.e., cluster) score and the threshold score for the endophenotype (i.e., cut-off score).

#### 4.5.3. Step 3: Assessment of BAP Endophenotypes in Family A and B

A team member who had not been involved in the phenotyping of Family A and B (Step 1) performed the endophenotype analysis (KT). Proportional scores for the four endophenotypes were calculated, and family members classified as having the endophenotype if their proportional score was greater than or equal to the cut-off scores identified in the small families analysis (Step 2). As above, the highest score (observed endophenotype score—threshold endophenotype score) for any endophenotype was used to represent an individual’s most prominent endophenotype. A discriminant function analysis was then used to determine the sensitivity and specificity of the endophenotype approach to identifying the presence of the BAP in these families. In addition, endophenotype results were correlated with measures of intellect, executive, social and adaptive functions using conservative non-parametric Spearman’s correlations (*r_s_*). Materials used for assessment and scoring of endophenotypes are available in Appendix A.

## 5. Conclusions

Despite significant advances towards unravelling the genetic heterogeneity of ASD, the underlying genetic aetiology remains unsolved for the majority of cases, in part due to significant challenges in identifying endophenotypes and potential carriers. We used a rigorous phenotyping approach to characterise the BAP in two large multiplex families with dominant inheritance of ASD and the BAP. Deep phenotyping identified four endophenotypes, showing differentiation of BAP features beyond traditional domain approaches. The proposed endophenotype approach advances current understanding and characterisation of the phenotypic spectrum for improved detection of the BAP that may facilitate gene discovery.

## Figures and Tables

**Figure 1 ijms-21-07965-f001:**
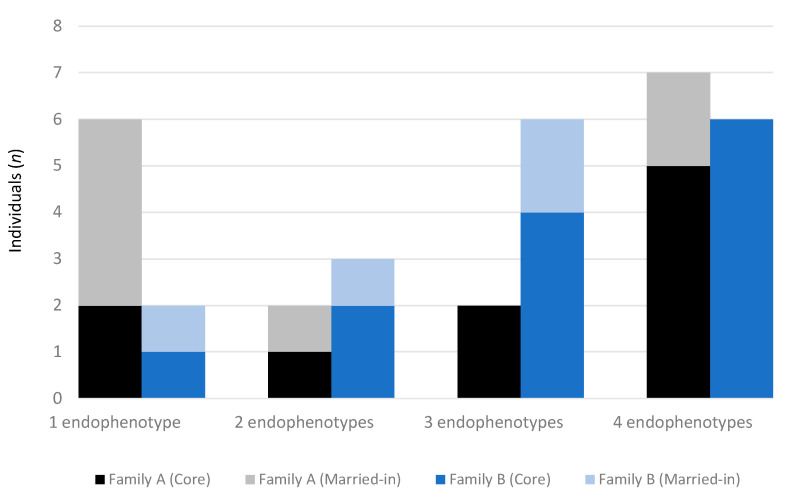
Number of BAP endophenotypes present in Family A and B. Individuals married-in to Family A tend to have a single endophenotype indicating a milder BAP presentation, in contrast with core family members who have multiple endophenotypes (obsessive most frequent). In Family B, married-in and core family members tend to have more than one endophenotype, with the aloof endophenotype the most frequent.

**Figure 2 ijms-21-07965-f002:**
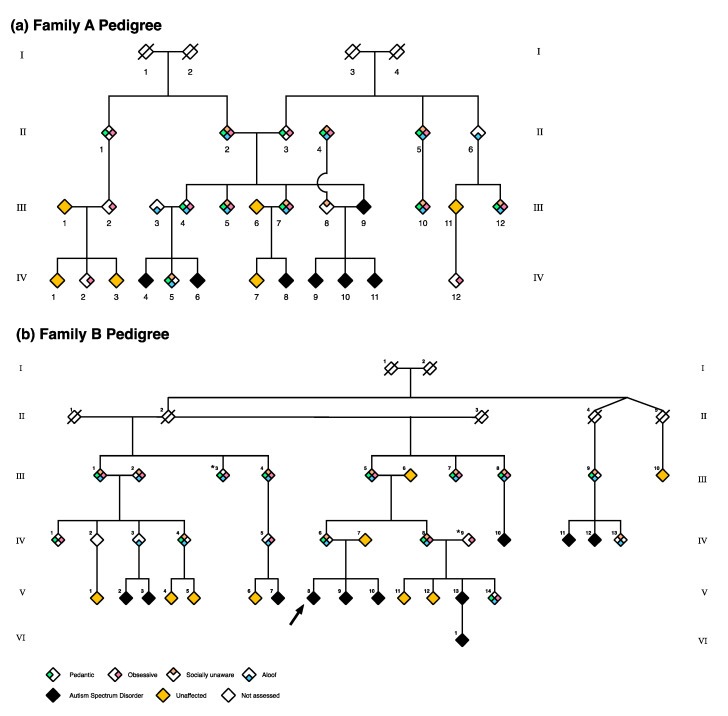
Scrambled pedigrees for Family A (panel **a**) and Family B (panel **b**) showing phenotypes and endophenotypes. All individuals with ≥1 endophenotype had the BAP, with the exception of two individuals from Family B (III-3 and IV-9) marked with an asterisk. These two individuals were clinically determined as unaffected but had above threshold endophenotype scores based on ROC curves. Some family members who were not phenotyped are not shown to preserve the anonymity of these families. The arrow indicates the proband shown in this pedigree. *: “All individuals with ≥1 endophenotype had the BAP, with the exception of two individuals from Family B (III-3 and IV-9) marked with an asterisk. These two individuals were clinically determined as unaffected but had above threshold endophenotype scores based on ROC curves.”.

**Figure 3 ijms-21-07965-f003:**
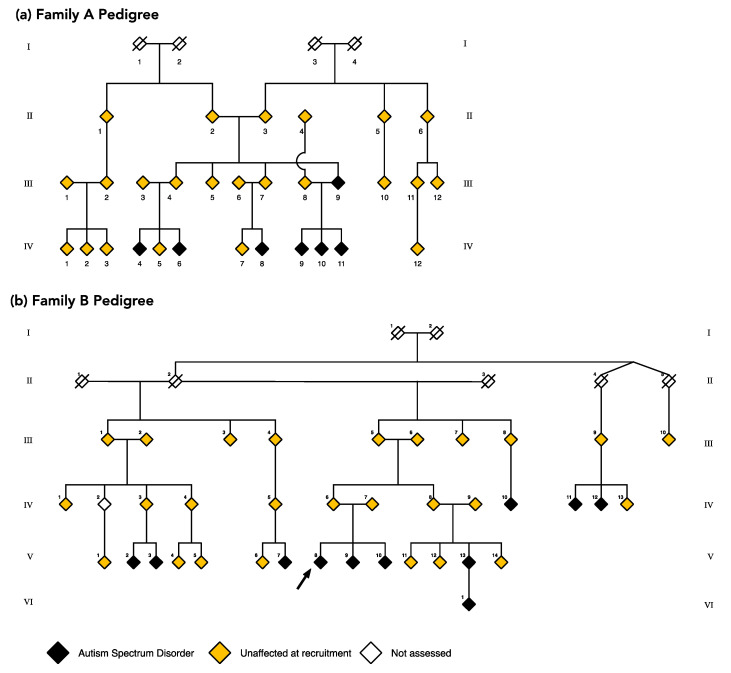
Scrambled pedigrees for Family A (Panel **a**) and Family B (Panel **b**) at recruitment. Individuals with a diagnosis of ASD are marked in black, and individuals recruited from the broader families are marked in yellow. White diamonds are individuals who were not assessed but are represented here to preserve the pedigree lines. One individual in Family B with a diagnosis of ASD was deceased (not shown to preserve anonymity) and one was too young to be assessed. The arrow indicates the proband shown in this pedigree.

**Figure 4 ijms-21-07965-f004:**
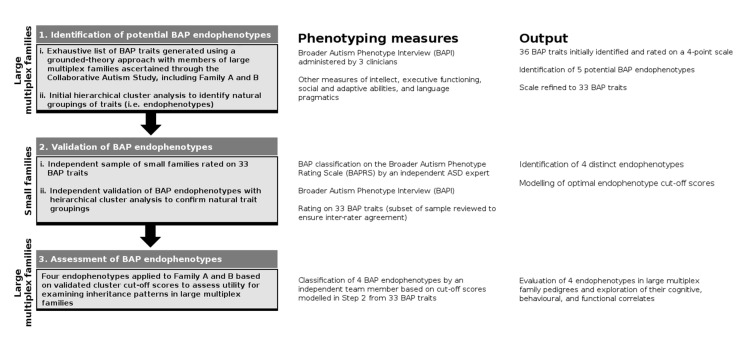
Iterative process used to identify and assess BAP endophenotypes. Step 1: An exhaustive list of BAP traits was generated through detailed clinical assessment of a number of large multiplex families, resulting in the identification of a final set of 33 BAP traits. Step 2. Validation analyses in an independent and aggregated sample of small families resulted in a 4-cluster solution representing distinct BAP endophenotypes. Step 3. Cut-off scores modelled in Step 2 were independently applied to large multiplex Family A and B for endophenotype classification and assessment of inheritance patterns.

**Table 1 ijms-21-07965-t001:** Intellectual functioning in Family A and B by diagnostic classification.

	Participants (*n*) (Female)	Mean Age (Range)	Cognitive Data (*n*)	FSIQ Mean (SD)	VIQ Mean (SD)	PIQ Mean (SD)
Family A						
ASD	7 (1)	11.43 (4–34)	5	107 (16)	97 (15)	101 (30)
Unaffected	6 (5)	25.83 (2–50)	5	127 (17)	130 (17)	117 (15)
BAP	17 (8)	49.18 (13–79)	16 ^a^	119 (13)	117 (13)	115 (12)
Total	30 (14)	35.7 (2–79)	26	118 (15)	116 (17)	113 (18)
Family B						
ASD	9 (3)	15.00 (8–20)	5	95 (20)	89 (28)	105 (16)
Unaffected	9 (7)	37.33 (10–73)	9	110 (14)	111 (14)	108 (11)
BAP	15 (10)	47.40 (15–73)	15	102 (17)	105 (18)	99 (17)
Total	33 (20)	35.06 (6–73)	29	103 (17)	104 (19)	102 (15)
Total	63 (34)	35.37 (2–79)	55 ^a^	110 (18)	110 (19)	107 (17)

FSIQ = Full Scale Intelligence Quotient, VIQ = Verbal Intelligence Quotient, PIQ = Performance Intelligence Quotient. Note. Average FSIQ = 80–119; Superior FSIQ = ≥120. ^a^ One individual only completed VIQ and select executive functioning subtests.

**Table 2 ijms-21-07965-t002:** Four endophenotypes of the BAP in the small families sample.

Mean Proportional Score (SD)		Cut-off Score	BAP Traits
Unaffected (*n =* 11)	BAP (*n* = 30)		
‘Socially unaware’: *Poor self-regulation and reciprocity in conversation*			
0.20 (0.18)	0.69 (0.06) **	>0.17	Reduced capacity for clear narrative Difficulty answering open ended questions Making inappropriate or awkward comments either on history or during assessments Tangential pragmatic style Tendency to monologue rather than participate in reciprocal conversation Tendency to anger easily Reduced quantity of verbal output
‘Pedantic’: *Self-focused and technical in interactions*			
0.04 (0.05)	0.11 (0.12) *	>0.14	An unusual or awkward greeting style Unusual eye gaze Speech has little variation in tone (i.e., monotonous) Unusual speech volume Precise articulation and language Terse pragmatic style Overly technical language Narcissistic personality style Focus on technicalities or minutiae Fastidious regarding personal appearance Self perception incongruent with views of others
‘Aloof’: *Difficulties relating to other’s emotions and expressing own emotions*			
0.12 (0.07)	0.31 (0.16) ***	>0.20	Aloof personality style Difficult or limited interpersonal relationships Reduced emotional empathy A limited capacity to develop rapport with assessors Reduced affection Awkward social interactions Opinionated in conversation Reduced cognitive empathy Little appreciation of humour (during the Cartoon task)
‘Obsessive’: *Regimented approach to life and tendency to ruminate*			
0.13 (0.10)	0.27 (0.18) ***	>0.25	Hobby or interest of unusual intensity, or restricted range of interests relative to peers Large collections or hoarding of items Fastidious cleaning Preference for structure in activities of daily living Recurrent thoughts that are not distressing Excessive worry

* *p* < 0.05, ** *p* < 0.01, *** *p* < 0.001.

**Table 3 ijms-21-07965-t003:** Correlations between endophenotypes and quantitative measures in Families A and B.

		Endophenotypes				
Domain	Task	Socially Unaware	Pedantic	Aloof	Obsessive	Total Number
Social communication	PRS	0.83 **	0.73 **	0.76 **	0.45 **	0.86 **
	FPT	−0.43 **	−0.28	−0.24	−0.18	−0.40 **
Intellect	FSIQ	−0.36 *	−0.10	−0.31 *	−0.02	−0.28
	VIQ	−0.29	−0.05	−0.31 *	−0.03	−0.28
	PIQ	−0.36 *	−0.13	−0.19	−0.02	−0.26
Executive functions	Trails (numbers) ^a^	−0.32 *	−0.20	−0.29	0.17	−0.19
	Trails (switch) ^a^	−0.24	−0.23	−0.27	−0.22	−0.34 *
	Design fluency (switch) ^a^	−0.27	−0.25	−0.15	−0.09	−0.34 *
	Design fluency (composite) ^a^	−0.27	−0.25	−0.25	−0.03	−0.32 *
	Tower task (achievement) ^a^	−0.40 **	−0.19	−0.27	−0.22	−0.26
	Sorting (confirmed)	−0.33 *	−0.23	−0.32 *	−0.22	−0.33 *
	Sorting (free sort)	−0.31	−0.15	−0.24	−0.37*	−0.37 *
Adaptive function	Social index (self-report)	−0.46 *	−0.35	−0.19	−0.31	−0.43 *

* *p* < 0.05; ** *p* < 0.01. PRS = Pragmatic Rating Scale, FPT = Faux Pas Test, FSIQ = Full Scale Intelligence Quotient, VIQ = Verbal Intelligence Quotient, PIQ = Performance Intelligence Quotient. ^a^ Nonverbal executive function subtests.

**Table 4 ijms-21-07965-t004:** Summary of the BAP endophenotypes and their functional correlates.

Endophenotype	Core Characteristic	Associated Functional Domains			
		Social	Intellect	Executive	Adaptive
Socially unaware	Poor self-regulation and reciprocity in conversation	✓	✓	✓	✓
Pedantic	Self-focused and technical in interactions	✓			
Aloof	Difficulties expressing and relating to other’s emotions	✓	✓	✓	
Obsessive	Regimented approach to life and tendency to ruminate	✓		✓	

**Table 5 ijms-21-07965-t005:** Protocol for diagnosing ASD and phenotyping the Broader Autism Phenotype (BAP) in large multiplex families.

	Participants with ASD		Participants without ASD		
Protocol Item	Child or Adolescent ≥ 4.5–17 yr	Adult ≥ 18 yr	Child < 13 yr	Adolescent ≥ 13–17 yr	Adult ≥ 18 yr
ADI-R + ADOS-G *or* DSM-IV interview + ADOS-G	+	±	−	−	−
Detailed developmental, medical, psychiatric and behavioural history	+	+	+	+	+
Family History Interview	−	+	−	−	±
Standardised testing of cognition and executive function ^a^	±	+	+	+	±
Questionnaires of adaptive behaviour ^b^	+	+	+	+	±
Broader Autism Phenotype Interview, the Faux Pas Task, Cartoon Task and Pragmatic Rating Scale	−	±	−	+	+
Physical Examination	+	+	+	+	+
High resolution molecular karyotype, Fragile X testing, metabolic investigations	+	−	−	−	−

ASD = Autism Spectrum Disorder, ADI-R = Autism Diagnostic Interview-Revised, ADOS-G = Autism Diagnostic Observation Schedule-Generic, DSM-IV = Diagnostic and Statistical Manual of Mental Disorders (4th edition). + all individuals completed this assessment, ± only some individuals completed this assessment, − no individuals completed this assessment. ^a^ Wechsler Abbreviated Scale of Intelligence and subtests of the Delis–Kaplan Executive Function System. ^b^ The Adaptive Behavioural Assessment System (2nd edition) and The Behavioural Rating Inventory of Executive Function.

**Table 6 ijms-21-07965-t006:** Demographics of the small families sample.

	Unaffected	BAP
Number of participants (female)	11 (6)	30 (19)
Mean age (range)	41.09 (18–53)	39.50 (14–53)
Mean BAPQ (SD) ^a^	2.43 (0.38)	2.92 (0.92)
Mean FSIQ (SD) ^b^	108 (14)	111 (13)
Mean VCI (SD) ^b^	106 (18)	109 (15)
Mean PRI (SD) ^b^	109 (7)	110 (15)

BAPQ = Broader Autism Phenotype Questionnaire, FSIQ = Full Scale Intelligence Quotient, VCI = Verbal Comprehension Index, PRI = Perceptual Reasoning Index. ^a^ Data available for unaffected (*n* = 9) and BAP (*n* = 18). ^b^ Data available for unaffected (*n* = 10) and BAP (*n* = 21).

## Data Availability

All data is available within the manuscript and its Supporting Information, although scrambled pedigrees have been used to preserve participant anonymity.

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
