# Peer review of "Tracing Autism Traits in Large Multiplex Families to Identify Endophenotypes of the Broader Autism Phenotype"

_ijms, 2020, doi:10.3390/ijms21217965_

Round 1

Reviewer 1 Report

This article is focused on the neurogenetics of ASD, and how this is associated with endophenotypes. Endophenotypes of ASD are of extreme interest nowadays, as they can help in the identification and treatment of ASD patients. Hence, this article is of great interest.

The article is very well written, analyzing a nice cohort of unique genetic cases, and analyze the data properly. The discussion is balanced and of high interest, and the overall impression from the article is very positive.

I have only few suggestions to improve the article:

In the introduction, authors should cite more studies to support there claims. 

For example, when mentioning social interaction difficulties in ASD, they should cite PMID: 29323671. When mentioning ASD genetics, they should cite PMID: 28941239.

It can be good to further discuss the polygenic architecture and de novo mutations associated with the majority of ASD cases, as mentioned by the authors, to educate more the reader on that matter. 

The numbers on figure 1 are not readable when the article is printed. Please re-edit. 

What is the purpose of the arrow in figure 1b and 4b? Please clarify

In the discussion, please elaborate on the clinical needs for endophenotypes in ASD, given the complexity involved in ASD.

Author Response

Thank you for your comments and suggestions. Please find attached our response and updated manuscript draft.

Reviewer 1 Report

This article is focused on the neurogenetics of ASD, and how this is associated with endophenotypes. Endophenotypes of ASD are of extreme interest nowadays, as they can help in the identification and treatment of ASD patients. Hence, this article is of great interest.The article is very well written, analyzing a nice cohort of unique genetic cases, and analyze the data properly. The discussion is balanced and of high interest, and the overall impression from the article is very positive.I have only few suggestions to improve the article:

C1.1 In the introduction, authors should cite more studies to support there claims. For example, when mentioning social interaction difficulties in ASD, they should cite PMID: 29323671. When mentioning ASD genetics, they should cite PMID: 28941239.

R1.1 Thank you for your insightful comments and suggestions. We have now included the recommended references in our updated literature review.

C1.2 It can be good to further discuss the polygenic architecture and de novo mutations associated with the majority of ASD cases, as mentioned by the authors, to educate more the reader on that matter. 

R1.2 Thank you for this suggestion. We have included a more nuanced discussion of this topic in the Introduction, which now reads:

“The recognition of milder phenotypes has facilitated an increased understanding and interest in the wide spectrum of clinical presentations in ASD. Family members of individuals with ASD have been observed to express milder forms, known as the broader autism phenotype (BAP), that are not sufficient to meet diagnostic criteria for ASD. The BAP includes a range of subtle behavioural and cognitive features that have qualitatively similar presentations to the core domains of ASD. While clinically significant impairment in these areas of functioning is seen in ASD, BAP traits are continuously distributed in the general population. Several studies have reported the expression of at least one BAP trait in 20% of relatives of children with ASD, as well as a higher rate of BAP traits compared to controls across domains of pragmatic language, personality, social cognition and executive function. Monozygotic twins demonstrate 65-90% concordance for ASD with a higher estimate when the BAP is considered, while dizygotic twin concordance is ~20%. Together these findings suggest a strong genetic basis for ASD. An estimated 25% of ASD cases can be identified clinically or molecularly with a predominant monogenic cause, with or without environmental factors. Although the aetiology for the majority of cases is still unknown, increasing evidence suggests a polygenic basis due to the interaction of multiple genetic risk variants following complex inheritance. Complementary techniques will be necessary to investigate the genetic aetiology of ASD in unsolved cases.”

C1.3 The numbers on figure 1 are not readable when the article is printed. Please re-edit. 

R1.3 We have uploaded the figure with original resolution, which should look correct in the full manuscript file, but may appear distorted in the proof preview document.

C1.4 What is the purpose of the arrow in figure 1b and 4b? Please clarify.

R1.4 The arrow is a conventional symbol indicating the proband in a genetic pedigree. We have now clarified this in Figures 1 and 4. Thank you for the suggestion.

C1.5 In the discussion, please elaborate on the clinical needs for endophenotypes in ASD, given the complexity involved in ASD.

R1.5 Thank you for raising this point. We have included a more detailed discussion of the merits of the endophenotype approach, which we put forward to be most useful for the identification of putative markers of genetic liability for ASD in family members of individuals with ASD, in the context of our study. We have updated our Introduction to reflect these points, which now reads:

“Endophenotypes are measurable features within a disorder that are proposed to reduce its complexity into more quantifiable elements. These components can be represented at any level of analysis, including but not limited to biochemical, neuroanatomical, neurophysiological, cognitive or neuropsychological measurements. They have been hypothesised to reflect more aetiologically homogeneous subgroups within genetically heterogeneous conditions. Endophenotypes are also presumed to be located closer to aetiological mechanisms in the pathway between genotype and disease, compared to more overt phenotypes that are used to define clinical syndromes. Clinically, the use of endophenotypes offers increased statistical power to localise and identify genes associated with disease. Importantly, an endophenotype must indicate genetic susceptibility to disease independent of disease status, and by definition may serve as a marker of genetic liability in individuals without the disorder. In individuals with the BAP, the mild expression of ASD-related traits is hypothesised to be due to increased genetic liability for ASD. There are several BAP traits that may be considered “endophenotypes” from within the domains of language, executive function, and social cognition. In the context of a single large family where numerous individuals demonstrate ASD or the BAP, recognition of BAP endophenotypes should allow granular identification of autism genes of dominant effect. This study is the first known to the authors to apply this approach in autism.”

Reviewer 2 Report

The proposed study addressed a very important topic in ASD in a  way that appears quite innovative.

My main concern is that the four endophenotypes proposed in the study fall a bit far from my understanding of an endophenotype (i.e., a disease-associated phenotype that is thought to directly express the neurobiological mechanisms underlying the more overt features of the disorder). I have been especially wondering whether the proposed endophenotypes meet the requirements they should obey (1. stable and reproducible; 2. reflecting a function of a certain brain system that determines the human behavior; 3. inherited). Despite I am aware that psychiatric endophenotypes are often based on articulated biobehavioral dimensions, I feel that the four endophenotypes proposed are quite complex and unlikely to be easily reproducible and, so that, they might end failing the objective of helping to uncover an association with specific genetic variants. On the other hand, 1) the procedure used to obtain the four endophenotypes appears to be rigorously base on their heritability, and 2) the evaluation made by three independent raters is intended to safeguard the reproducibility of the endophenotypes. 

This said, I have three suggestions:

  1. I think it would be important to guarantee that the endophenotypes are also stable over time, besides across different raters. Did the subjects were retested later or the three test included were made in different times? Do authors have any chance to show that the endophenotypes are stable irrespective of current state, comorbid disorders and so on?
  2. Secondly, once reproducibility has been shown, It will also be important that authors enable other researchers to use their putative endophenotypes in their future studies. To this aim, a practice guideline could be provided as supplementary material to allow researchers to find and measure the four endophenotypes proposed.
  3. Eventually, since the four endophenotypes actually sound like phenotypes (aloof, obsessive..), I think It may be important to carefully warn about these profiles being intermediate corps between genotype and phenotypes and not meant to be used merely as clinical descriptions or to validate the diagnosis.

Author Response

(The authors gave the same response as above.)

Reviewer 2 Report

C2.1 The proposed study addressed a very important topic in ASD in a way that appears quite innovative. My main concern is that the four endophenotypes proposed in the study fall a bit far from my understanding of an endophenotype (i.e., a disease-associated phenotype that is thought to directly express the neurobiological mechanisms underlying the more overt features of the disorder). I have been especially wondering whether the proposed endophenotypes meet the requirements they should obey (1. stable and reproducible; 2. reflecting a function of a certain brain system that determines the human behavior; 3. inherited). Despite I am aware that psychiatric endophenotypes are often based on articulated biobehavioral dimensions, I feel that the four endophenotypes proposed are quite complex and unlikely to be easily reproducible and, so that, they might end failing the objective of helping to uncover an association with specific genetic variants. On the other hand, 1) the procedure used to obtain the four endophenotypes appears to be rigorously base on their heritability, and 2) the evaluation made by three independent raters is intended to safeguard the reproducibility of the endophenotypes

R2.1 Thank you for raising this excellent point and for your thoughtful comments. We refer specifically to Gottesman and Gould’s (2003) seminal criteria for an endophenotype, as well as revisions proposed by Beauchaine and Constantino (2017), which have shaped the concept of an endophenotype over the last two decades and constitute the widely accepted definition in the field. We argue that our proposed endophenotypes fulfill Gottesman and Gould’s (2003) criteria, as described in Beauchaine and Constantino (2017), which state that endophenotypes must:

  • Segregate with illness in the general population
  • Be heritable
  • Be state independent, manifesting whether illness is present or in remission.
  • Cosegregate with the disorder within families.
  • Be present at higher rates within affected families than in the general population.
  • Be characteristics that can be measured reliably, and specific to the illness of interest.

There are two exceptions to this definition that we acknowledge. The first is the criterion of specificity to a disorder, as the ASD-related traits that emerge as BAP endophenotypes in our study may also be observable in other psychiatric conditions. Beauchaine and Constantino (2017) have pointed out that the disorder-specificity criterion is problematic given the strong overlap in the complex aetiology of psychiatric disorders and genetic liability potentially underlying a spectrum of trait expression. They have therefore proposed that this criterion be relaxed and argued that endophenotypes indicating vulnerability to multiple disorders may be highly informative. We have now included this point in our discussion, which reads:

“Here we delineated the BAP into distinct endophenotypes that fulfil Gottesman and Gould’s criteria for a true endophenotype (Supplementary Table S2). This includes recent proposed revisions to account for the strong overlap in the complex aetiology and genetic liability underlying the spectrum of trait expression across many neurodevelopmental and psychiatric conditions, with the expectation that putative endophenotypes may not be strictly disorder-specific.”

The second exception relates to the rate of the proposed endophenotypes being higher in affected families than in the general population. While it was beyond the scope of our study to investigate this in unrelated controls, we have achieved our aim of characterising distinct endophenotypes in two independent samples that show high sensitivity (97%) and specificity (82%) to individuals independently classified with the BAP. Moreover, previous research into the BAP has already established that it occurs at a higher rate in relatives of probands with ASD compared to the general population. In response to the Reviewer’s concerns, however, we have now acknowledged this as a limitation in our Discussion, which reads:

“Successful gene identification in future work requires capture of all individuals who may carry the putative variant, with the approach outlined here designed to enable more robust gene identification work. Future genetic investigations are required to test the reproducibility of the four identified endophenotypes over time and in additional independent samples, as well as determine their rate of occurrence in affected families compared to the general population (see Supplementary materials for guidelines and measures of the four endophenotypes).”

Please also see R2.4 where we argue that endophenotypes are agnostic to the level of analysis (i.e., may be represented by neuropsychological measurement).

C2.2 This said, I have three suggestions. I think it would be important to guarantee that the endophenotypes are also stable over time, besides across different raters. Did the subjects were retested later or the three test included were made in different times? Do authors have any chance to show that the endophenotypes are stable irrespective of current state, comorbid disorders and so on?

R2.2 While we have established inter-rater reliability in our study, investigating test-retest reliability through longitudinal assessment was beyond the scope of the current work. Consequently, in the Discussion we have acknowledged this as a limitation that requires future investigation, as noted in our response for R2.1 above.

C2.3 Secondly, once reproducibility has been shown, It will also be important that authors enable other researchers to use their putative endophenotypes in their future studies. To this aim, a practice guideline could be provided as supplementary material to allow researchers to find and measure the four endophenotypes proposed.

R2.3 Thank you for this suggestion. We have now included guidelines and assessment materials in our Supplementary materials to support replication of our findings in future studies. We have also included signposts in our Methods and Discussion to guide the reader to these resources (see R2.1 above).

C2.4 Eventually, since the four endophenotypes actually sound like phenotypes (aloof, obsessive..), I think It may be important to carefully warn about these profiles being intermediate corps between genotype and phenotypes and not meant to be used merely as clinical descriptions or to validate the diagnosis.

R2.4 Thank you for raising this excellent point. Here we differentiate phenotypes (core symptom domains of ASD) from endophenotypes as subclinical trait expression associated with an underlying genetic liability for ASD (Lenzenweger 2013). We agree that this is important to clarify for the reader and have now added text to the Introduction to address this in more detail, which reads:

“Endophenotypes are measurable features within a disorder that are proposed to reduce its complexity into more quantifiable elements. These components can be represented at any level of analysis, including but not limited to biochemical, neuroanatomical, neurophysiological, cognitive or neuropsychological measurements. They have been hypothesised to reflect more aetiologically homogeneous subgroups within genetically heterogeneous conditions. Endophenotypes are also presumed to be located closer to aetiological mechanisms in the pathway between genotype and disease, compared to more overt phenotypes that are used to define clinical syndromes. Clinically, the use of endophenotypes offers increased statistical power to localise and identify genes associated with disease. Importantly, an endophenotype must indicate genetic susceptibility to disease independent of disease status, and by definition may serve as a marker of genetic liability in individuals without the disorder. In individuals with the BAP, the mild expression of ASD-related traits is hypothesised to be due to increased genetic liability for ASD. There are several BAP traits that may be considered “endophenotypes” from within the domains of language, executive function, and social cognition. In the context of a single large family where numerous individuals demonstrate ASD or the BAP, recognition of BAP endophenotypes should allow granular identification of autism genes of dominant effect. This study is the first known to the authors to apply this approach in autism.”